

# Using an antidiffusive transport scheme in the vertical direction : a promising novelty for chemistry-transport models

Sylvain Mailler[1,2], Romain Pennel[1], Laurent Menut[1], and Mathieu Lachâtre[1]

[1]LMD/IPSL, École Polytechnique, Institut Polytechnique de Paris, ENS, PSL Research University, Sorbonne Université, CNRS, Palaiseau France
[2]École des Ponts-ParisTech, Marne-la-Vallée, France

**Correspondence:** Sylvain Mailler (sylvain.mailler@lmd.polytechnique.fr)

**Abstract.** The potential interest of the antidiffusive transport scheme proposed by Després and Lagoutière (1999) for resolving vertical transport in chemistry-transport models is investigated in an idealized framework with very encouraging results. We show that, compared to classical higher-order schemes, the Després and Lagoutière (1999) scheme reduces numerical diffusion and improves accuracy in idealized cases that are typical of atmospheric transport of tracers in chemistry-transport models. Increased accuracy and reduced diffusion is spectacular when, and only when vertical resolution is insufficient to properly resolve vertical gradients, which is very frequent in chemistry-transport models. Therefore, we think that this scheme is an extremely promising solution for reducing numerical diffusion in chemistry-transport models.

## 1 Introduction

Reducing numerical errors in chemistry-transport models (CTMs) is an important direction into improvement of these models. Among the well-known errors in eulerian CTMs, excessive numerical diffusion in all directions is a well-known drawback many such models, and in the past past decade, excessive or poorly represented vertical transport and diffusion has been identified as a major cause for numerical dispersion in these models (Vuolo et al., 2009; Emery et al., 2011; Mailler et al., 2017). This excessive vertical diffusion may have a strong impact on representation of ground-level ozone concentrations due to spurious transport of stratospheric ozone into the troposphere (Emery et al., 2011), and also hinders the ability of eulerian CTMs to represent accurately intercontinental transport of dense polluted plumes such as volcanic plumes (Colette et al., 2010; Mailler et al., 2017; Lachatre et al., 2020). While CTMs manage to represent such plumes in terms of general location, they typically fail to maintain the fine-scale structure of the plumes, and tend to dilute them too much observations (Colette et al., 2010; Mailler et al., 2017).

Among other more model-specific solutions, Emery et al. (2011) suggest as the main paths to solve this problem increasing vertical resolution and improving the vertical advection scheme, in their case switching from a first-order to a second-order advection scheme. In the same line, Eastham and Jacob (2017); Zhuang et al. (2018) have discussed the need for increased vertical resolution in order to adequately represent long-range advection of chemical plumes.

In the line of improving the vertical advection scheme, Lachatre et al. (2020) describes the implementation of the Després and Lagoutière (1999) antidiffusive transport scheme for vertical transport in the CHIMERE chemistry-transport model (Mailler





et al. (2017)) and its application to modelling the March 18, 2012 eruption of Mount Etna (Italy). They show that using this transport scheme reduces numerical diffusion and permits a better representation of the volcanic plume after long-range advection, thereby showing that it is possible to strongly reduce numerical diffusion in CTMs without increasing the number of vertical levels. However, that study, set in the framefork of a full-fledge chemistry-transport model fed by real-life atmospheric fields makes it difficult to fully disentangle the effect of the transport scheme itself from other effects such as uncertainties on

emission fluxes and mass-wind inconsistencies in the forcing meteorological fields.

Therefore, the present study aims at answering the questions on the use of Després and Lagoutière (1999) that could not be adressed in the realistic framework of Lachatre et al. (2020). For that purpose, we have designed three idealized test cases that permit to compare the performance of the Després and Lagoutière (1999) scheme with the classical schemes of Van Leer (1977) and Colella and Woodward (1984), not only in terms of diffusion but also in terms of accuracy compared to the exact

solution, which could not be done in Lachatre et al. (2020). We also examine how the performance of Després and Lagoutière (1999) compares to the two above-cited schemes and to the order-1 upwind Godunov scheme when resolution increases.

Section 2 describes the numerical methods that have been used, their implementation, the discretization strategies and the cases that have been designed for the study. Section 3 presents simulation outputs and the diagnostics that have been designed to compare the different transport strategies with each other. Section 4 discusses the results, and section 5 gives our conclusions.

## 40 2 Numerical methods and case description

Continuity equation for the motion of air is as follows:

$$\frac{\partial C}{\partial t} + \nabla\left(\mathbf{\Phi}\right) = 0, \tag{1}$$

where $u$ represents wind speed, $C$ is air concentration (all species together) in molecules per unit volume and $\mathbf{\Phi} = C\mathbf{u}$ represent the air flux vector.

The continuity equation for species $s$ is as follows:

$$\frac{\partial C_s}{\partial t} + \nabla\left(\mathbf{\Phi}_s\right) = 0, \tag{2}$$

where $C_s$ is concentration of species $s$ in molecules per unit volume and $\mathbf{\Phi}_s = C_s\mathbf{u}$ is the flux vector for species $s$. Equivalently, Eq. 2 becomes:

$$\frac{\partial C_s}{\partial t} + \nabla\left(\alpha_s\mathbf{\Phi}\right) = 0, \tag{3}$$

where $\alpha_s$ is the mixing ratio for species $s$:

$$\alpha_s = \frac{C_s}{C}. \tag{4}$$





Chemistry-transport models try to solve Eq. 3 as accurately as possible on their discretized grids, while keeping the numerical cost of numerical resolution under control.

### 2.1 1d discretization of advection and advection schemes

In 1d, Eq. 2 becomes:

$$\frac{\partial C_s}{\partial t} + \frac{\partial (C_s \mathbf{u})}{\partial x} = 0, \tag{5}$$

Here we follow a semi-lagrangian approach by identifying the air parcels that have entered cell $i$ through its left boundary, and conversely the air parcels that have left cell $i$ through its right boundary (we suppose that the wind is positive). Let $\Delta^- x$ be so that the Lagrangian trajectory starting at $x_{i-\frac{1}{2}} - \Delta^- x$ at time $t$ passes through $x_{i-\frac{1}{2}}$ at time $t + \Delta t$. $\Delta^- x$ is the distance

travelled by the last air particle entering grid cell $i$ at time $t + \Delta t$. Let us define the wind speed representative of facet $i - \frac{1}{2}$ between $t$ and $t + \Delta t$ as $\overline{u}_{i-\frac{1}{2}} = \frac{\Delta^- x}{\Delta t}$. If we define the wind speed representative of right facet $\overline{u}_{i+\frac{1}{2}}$ in a similar way as $\frac{\Delta^+ x}{\Delta t}$ where $\Delta^+ x$ is so that the Lagrangian trajectory starting at $x_{i+\frac{1}{2}} - \Delta^+ x$ at time $t$ passes through $x_{i+\frac{1}{2}}$ at time $t + \Delta t$, then Eq. 5 can be discretized over cell $i$ as:

$$\frac{C_{s,i}(t + \Delta t) - C_{s,i}(t)}{\Delta t} = \frac{\overline{u}_{i-\frac{1}{2}} \overline{C}_{s,i-\frac{1}{2}} - \overline{u}_{i+\frac{1}{2}} \overline{C}_{s,i+\frac{1}{2}}}{x_{i+\frac{1}{2}} - x_{i-\frac{1}{2}}} \tag{6}$$

where

$$\overline{C}_{s,i-\frac{1}{2}} = \frac{1}{\Delta^- x} \int_{x_{i-\frac{1}{2}} - \Delta^- x}^{x_{i-\frac{1}{2}}} C_s(x,t) \, \mathrm{d}x \tag{7}$$

and

$$\overline{C}_{s,i+\frac{1}{2}} = \frac{1}{\Delta^+ x} \int_{x_{i+\frac{1}{2}}}^{x_{i+\frac{1}{2}} + \Delta^+ x} C_s(x,t) \, \mathrm{d}x. \tag{8}$$

Eq. 6 is verified exactly with no particular hypothesis on the wind speed $u(x,t)$ nor the concentration field $C_s(x,t)$. For air

concentration, the continuity equation can be discretized in the same way:

$$\frac{C_i(t + \Delta t) - C_i(t)}{\Delta t} = \frac{\overline{u}_{i-\frac{1}{2}} \overline{C}_{i-\frac{1}{2}} - \overline{u}_{i+\frac{1}{2}} \overline{C}_{i+\frac{1}{2}}}{x_{i+\frac{1}{2}} - x_{i-\frac{1}{2}}} \tag{9}$$





In order to permit monotonicity of the advection scheme in terms of mixing ratios (i.e. the mixing ratio for species $s$ stays within its initial range) , we reformulate Eq. 6 by using fluxes and mixing ratios instead of winds and concentrations by introducing:

$$\overline{\alpha}_{s,i\pm\frac{1}{2}} = \frac{\overline{C}_{s,i\pm\frac{1}{2}}}{\overline{C}_{i\pm\frac{1}{2}}} \tag{10}$$

and

$$\overline{F}_{i\pm\frac{1}{2}} = \overline{C}_{i\pm\frac{1}{2}}\overline{u}_{s,i\pm\frac{1}{2}} \tag{11}$$

Then equations 6 and 9 become:

$$\frac{C_{s,i}\left(t+\Delta t\right)-C_{s,i}\left(t\right)}{\Delta t} = \frac{\overline{F}_{i-\frac{1}{2}}\overline{\alpha}_{s,i-\frac{1}{2}} - \overline{F}_{i+\frac{1}{2}}\overline{\alpha}_{s,i+\frac{1}{2}}}{x_{i+\frac{1}{2}} - x_{i-\frac{1}{2}}} \tag{12}$$

and

$$\frac{C_{i}\left(t+\Delta t\right)-C_{i}\left(t\right)}{\Delta t} = \frac{\overline{F}_{i-\frac{1}{2}} - \overline{F}_{i+\frac{1}{2}}}{x_{i+\frac{1}{2}} - x_{i-\frac{1}{2}}} \tag{13}$$

Equations 12-13 are a flux-form reformulation of semi-lagrangian equations 6-9. The form of Eqs. 12-13 makes it straightforward to verify that if Eq. 13 is verified and if $\overline{\alpha}_{s,i-\frac{1}{2}}$ lies between $\overline{\alpha}_{s,i-1}$ and $\overline{\alpha}_{s,i}$ (and if $\overline{\alpha}_{s,i+\frac{1}{2}}$ lies between $\overline{\alpha}_{s,i}$ and $\overline{\alpha}_{s,i+1}$) then the resulting advection scheme guarantees monotonicity of mixing ratios, which is physically desirable and is the

reason why chemistry-transport models usually resolve advection of trace species using an approach based on Eq. 12 rather than a straightforward resolution of Eq. 6.

Regarding chemistry-transport models, in practice, approximated values of $\overline{F}_{i-\frac{1}{2}}$ are inferred from the wind and density fields provided by the forcing meteorological dataset. Here we will avoid the typical problem of mass-wind inconsistencies-discussed in, e.g., Jöckel et al. (2001); Emery et al. (2011); Lachatre et al. (2020), by working with analytically-defined

non-divergent mass fluxes, and constant and uniform air density, so that Eq. 12 is verified exactly by construction.

This simplified framework will permit us to focus on the transport scheme of the chemistry-transport model whose task, in flux-form, is to estimate the values of $\overline{\alpha}_{s,i\pm\frac{1}{2}}$ that are needed for numerical resolution of Eq. 12.

### 2.1.1    Advection schemes and tracer flux calculation

**The Godunov donor-cell scheme**

The most simple way of estimating $\overline{\alpha}_{s,i\pm\frac{1}{2}}$ is the Godunov donor-cell scheme, simply evaluating $\overline{\alpha}_{s,i,k+\frac{1}{2}}$ as:





$$\overline{\alpha}_{s,i+\frac{1}{2}} = \alpha_{s,i} \quad \text{if } \overline{F}_{i+\frac{1}{2}} \geq 0 \tag{14}$$

$$\overline{\alpha}_{s,i+\frac{1}{2}} = \alpha_{s,i+1} \text{ if } \overline{F}_{i+\frac{1}{2}} < 0 \tag{15}$$

This order-1 scheme is cheap, robust and mass-conservative but extremely diffusive. It is therefore important to find more accurate ways to estimate $\overline{\alpha}_{s,i+\frac{1}{2}}$.

**The Van Leer (1977) scheme**

The second-order slope-limited scheme of Van Leer (1977) brought to our notations and assuming uniform air concentration yields the following expression of $\overline{\alpha}_{s,i+\frac{1}{2}}$ (for $\overline{F}_{i+\frac{1}{2}} > 0$).

$$\overline{\alpha}_{s,i+\frac{1}{2}} = \alpha_{s,i} + \frac{1-\nu}{2}\mathrm{sign}\left(\alpha_{s,i+1}-\alpha_{s,i}\right)\mathrm{Min}\left(\frac{1}{2}\left|\alpha_{s,i+1}-\alpha_{s,i-1}\right|, 2\left|\alpha_{s,i+1}-\alpha_{s,i}\right|, 2\left|\alpha_{s,i}-\alpha_{s,i-1}\right|\right), \tag{16}$$

where $\nu = \frac{\Delta^+ x}{x_{i+\frac{1}{2}}-x_{i-\frac{1}{2}}}$ is the Courant number for the donor cell. If $\nu > 1$, then more mass leaves the cell than the mass that was initially present and the Courant-Friedrichs-Lewy condition is violated, yielding numerical instability. If $\alpha_{s,i}$ is a local extremum of mixing ratio $((\alpha_{s,k}-\alpha_{s,k-1})(\alpha_{s,k+1}-\alpha_{s,k}) \leq 0)$, no interpolation is performed and $\overline{\alpha}_{s,i+\frac{1}{2}} = \alpha_{s,i}$ is imposed: in this case, the scheme falls back to the simple Godunov donor-cell formulation (Eq. 14). This order-2 scheme has been used for decades in chemistry-transport modelling, being a good tradeoff between reasonably weak diffusion, at least compared to more simple schemes such as the Godunov donor-cell scheme, and small computational burden compared to higher-order schemes such as the Piecewise Parabolic Method (Colella and Woodward, 1984).

**The Colella and Woodward (1984) Piecewise Parabolic Method**

The Colella and Woodward (1984) Piecewise Parabolic Method (PPM) consists in performing a parabolic reconstruction of the concentration field inside each model cell using information from three upwind cells and two downwind cells, and applying limiters to preserve the scheme's monotonicity and stability. The detailed procedure is described in the seminal Colella and Woodward (1984) paper. Our implementation of this method has been adapted from the CASTRO Compressible Astrophysical Solver Almgren et al. (2010). While third-order by design, application of limiters in the vicinity of extrema introduces first-order truncature errors in their vicinity so that third-order convergence is not expected with the PPM scheme as descibed in Colella and Woodward (1984) (Colella and Sekora, 2008). However, this limitation does not prevent the Colella and Woodward (1984) method to give much better results than simpler order-2 schemes such as Van Leer (1977), so that the Colella and Woodward (1984) PPM scheme has been used successfully for a wide range of applications including meteorological modelling (Carpenter et al., 1990), chemistry-transport modelling (Vuolo et al., 2009), astrophysics (Almgren et al., 2010) etc.

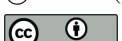



**The Després and Lagoutière (1999) scheme**

The scheme of Després and Lagoutière (1999) is defined by their equations 2 to 4. If $F_{i+\frac{1}{2}} > 0$, these equations brought to the notations of Eq. 12, give:

$$\overline{\alpha}_{s,i\frac{1}{2}} = \alpha_{s,k} + \frac{1-\nu}{2} \mathrm{Max}\left[0, \mathrm{Min}\left(\frac{2}{\nu}\frac{\alpha_{s,i} - \alpha_{s,i-1}}{\alpha_{s,i+1} - \alpha_{s,i}}, \frac{2}{1-\nu}\right)\right] \times (\alpha_{s,i+1} - \alpha_{s,i}), \tag{17}$$

with the same notations as for the Van Leer (1977) scheme (above). As above, if $((\alpha_{s,i} - \alpha_{s,i-1})(\alpha_{s,i+1} - \alpha_{s,i}) \leq 0)$, no interpolation is performed and the scheme falls back to the simple Godunov donor-cell formulation (Eq. 14). As stated by its authors, this scheme is antidiffusive. Unlike other schemes such as the Van Leer (1977) scheme described above, two unusual choices have been made by the authors in order to minimize diffusion by the advection scheme:

– Their scheme is accurate only to the first order

– The scheme is linearly unstable, but non-linearly stable (their Theorem 1)

The idea of the authors has been to make the interpolated value $\overline{\alpha}_{s,i+\frac{1}{2}}$ as close as possible to the downstream value ($\alpha_{s,i+1}$ if the flux is positively oriented). This property is desirable because it is the key property in order to reduce numerical diffusion as much as mathematically possible while still maintaining the scheme stability. The authors present 1d case-studies with their scheme obtaining extremely interesting results: fields that are initially concentrated on one single cell do not occupy more than 3 cells even after a long advection time (their Fig. 2), sharp gradients are very well preserved (their Fig. 1), and, more unexpectedly due to its antidiffusive character, the scheme also performs well in maintaining the shape of concentration fields with an initially smooth concentration gradient. After extensive testing, these authors also suggest (their Conjecture 1) that convergence of the simulated values towards exact values occur even if the time step is reduced before the space step: in simpler terms, this means that the scheme performs very well even at small CFL values, a property that is not shared by most advection schemes. Comparison of Eq. 17 with 16 shows that the numerical cost of the Després and Lagoutière (1999) scheme is about the same as the Van Leer (1977) scheme.

### 2.1.2 2d discretization of advection and directional splitting

All the simulations in the present study are 2d $x$-$z$ cases on a domain discretized over a regular cartesian mesh. In order to work with the usual units (concentrations per unit volume and not per unit area), a third, degenerate space dimension has been introduced, with one grid cell in the $y$ direction, $\delta y = \delta x$. Zero mass flux is prescribed in the $y$ direction. Index $i$ is attributed to the $x$ direction, index $k$ to the $z$ direction, and no index is attributed to the degenerate $y$ dimension. Eq. 12 becomes:





| Abbreviation/long name | horizontal transport | vertical transport | Splitting | Expected convergence order |
|---|---|---|---|---|
| Upwind | Colella and Woodward (1984) | Godunov | Lie | 1 |
| Van Leer | Colella and Woodward (1984) | Van Leer (1977) | Strang | 2 |
| PPM | Colella and Woodward (1984) | Colella and Woodward (1984) | Strang | 2 |
| DL99 | Colella and Woodward (1984) | Després and Lagoutière (1999) | Lie | 1 |

**Table 1.** Summary of the different transport configurations that have been tested

$$\frac{C_{s,i,k}\left(t+\Delta t\right)-C_{s,i,k}\left(t\right)}{\Delta t} = \frac{\overline{F}_{i-\frac{1}{2},k}\overline{\alpha}_{s,i-\frac{1}{2},k} - \overline{F}_{i+\frac{1}{2},k}\overline{\alpha}_{s,i+\frac{1}{2},k}}{x_{i+\frac{1}{2}} - x_{i-\frac{1}{2}}} \tag{18}$$

$$+ \frac{\overline{F}_{i,k-\frac{1}{2}}\overline{\alpha}_{s,i,k-\frac{1}{2}} - \overline{F}_{i,k+\frac{1}{2}}\overline{\alpha}_{s,i,k+\frac{1}{2}}}{z_{k+\frac{1}{2}} - z_{k-\frac{1}{2}}}$$

$$\tag{19}$$

Here, $\overline{F}_{i-\frac{1}{2},k}$ is the time-averaged mass flux of air through the left boundary of cell $i,k$ between $t$ and $t+\Delta t$, and similar definitions for $\overline{F}_{i+\frac{1}{2},k}$ and $\overline{F}_{i,k\pm\frac{1}{2}}$. $\overline{\alpha}_{s,i-\frac{1}{2},k}$ is the mixing ratio of species $s$ in the air volume entering cell $i,k$ through its left boundary between $t$ and $t+\Delta t$. If $\mathcal{V}$ is the geometric volume containing at time $t$ all the air parcels that are going to cross the left boundary of cell $i,k$ between $t$ and $t+\Delta t$ then:

$$\overline{\alpha}_{s,i-\frac{1}{2},k} = \frac{\int_{\mathcal{V}} C_s\left(x,z,t\right)\mathrm{d}\mathcal{V}}{\int_{\mathcal{V}} C\left(x,z,t\right)\mathrm{d}\mathcal{V}} \tag{20}$$

From a practical point of view, it is extremely difficult to actually find the countours of $\mathcal{V}$, reconstruct and integrate the concentrations of air and of tracer over this volume. This is why, in practice, Eulerian model in cartesian grids tend to split between the two (or three) space directions. Integrating separately direction $x$ and then direction $z$ over time $\Delta t$ (generally calles 'Lie splitting') gives order-1 error, and applying the so-called Strang splitting by integrating first in the $x$ direction over 160 $\frac{\Delta t}{2}$, then in the $z$ direction over $\Delta t$, and finally once again in the $x$ direction over $\frac{\Delta t}{2}$ reduces the splitting error to order-2.

### 2.1.3 Tested configurations

Since the present study is aimed at studying vertical transport only, we chose to test the Godunov, Van Leer (1977), Colella and Woodward (1984) and Després and Lagoutière (1999) schemes in the vertical direction with the same transport strategy over the $x$ axis, namely the Colella and Woodward (1984), our best available option. Lie splitting has been applied for scheme 165 combinations with expected order-1 accuracy, while Strang splitting has been applied for scheme combinations with expected order-2 accuracy. The configurations that have been tested are summarized in Table 1.





| nx | $\Delta x$ (m) | $L = nx \times \Delta x$ (km) | nz | $\Delta z$ (m) | Duration |
|---|---|---|---|---|---|
| | | Cases 1 and 2 | | | |
| 80 | 25 000 | 2 000 | 24 | 500 | $2T$ |
| | | Cases 3 and 4 (convergence tests) | | | |
| 20 | 50 000 | 1 000 | 12 | 1 000 | $T$ |
| 40 | 25 000 | 1 000 | 24 | 500 | $T$ |
| 80 | 12 500 | 1 000 | 48 | 250 | $T$ |
| 160 | 6 250 | 1 000 | 96 | 125 | $T$ |
| 320 | 3 125 | 1 000 | 192 | 62.5 | $T$ |

**Table 2.** Domain resolution and size for Cases 1 and 2; set of increasing resolutions used for convergence tests (Cases 3 and 4). For all configurations, $nz * \Delta z = 12\,000\,\mathrm{m}$ is the domain vertical extension

## 2.2 Test case definition

We have defined three test-cases designed to be representative of long-range tracer transport situations in the atmosphere. The simulation domain covers an $x$-$z$ domain with length $L = 2000\,\mathrm{km}$ from west to east and thickness $H = 12\,\mathrm{km}$, with a periodic

boundary conditions at the lateral boundaries, and open boundaries at the top and at the bottom of the simulation domain, with clean air (zero tracer concentration) entering from these boundaries. We will use $T = 86400\,\mathrm{s}$, the length of a complete day on Earth, as time scale for the case studies, along with the corresponding pulsation $\omega = \frac{2\pi}{T}$. The number density of the carrying fluid (air) will be assumed uniform. These simplifying assumptions are designed to ease the formalism and the formulation of exact solutions of the problem. Two situations of relevance for atmospheric tracer transport have been represented, along

with another numerical experiment designed in order to investigate the properties of the tested transport configurations in terms of convergence rate. Case 1, presented in 2.2.1 aims to represent the formation of a thin plume from an initially thicker tracer volume through the action of zonal wind shear, a situation typical of long-range advection of polluted plumes in the free troposphere. Case 2, presented 2.2.2 represents long-range advection of a thin plume under the action of a strong zonal wind.

Except for tests of convergence rates for which increasingly fine discretizations have to be tested, the domain is discretized

into 80 evenly-spaced cells from west to east ($\Delta x = 25\,\mathrm{km}$) and 24 evenly-spaced cells from bottom to top ($\Delta z = 500\,\mathrm{m}$).

### 2.2.1 Case 1: Thin layer formation under wind shear

In this case, we consider the evolution of an inert tracer initially distributed as follows:

$$\alpha\left(t=0,x,z\right) = \alpha_m \text{ if } \frac{H}{2} - h_1 \leq z \leq \frac{H}{2} + h_1 \text{ and } \frac{L}{2} - \delta x_1 \leq x \leq \frac{L}{2} + \delta x_1 \tag{21}$$

$$\alpha\left(t=0,x,z\right) = 0 \text{ otherwise,} \tag{22}$$





With $h_1 = 1500\,\mathrm{m}$ the half-thickness of the initial layer, $\delta x_1 = 25\,\mathrm{km}$ the half-length of the initial layer, H the height of domain top ($H = 12\,km$, see Tab. 2) and L the length of domain ($L = 2000\,\mathrm{km}$). This describes a uniform plume initially confined vertically between $z = 4500\,\mathrm{m}$ and $z = 7500\,\mathrm{m}$ and horizontally between $x = 975\,\mathrm{km}$ and $x = 1025\,\mathrm{km}$, with an initial (and arbitrary) mixing ratio of $100\,\mathrm{ppb}$.

Zonal wind is constant in time, zonally uniform and vertically sheared:

$$u(x,z,t) = U_0 \frac{2z}{H} \tag{23}$$

with $U_0 = \frac{L}{2T}$ so that the horizontal motion of fluid at $z = \frac{H}{2}$ brings it back at its initial position after a time $2T$, which will be the duration of the numerical experiment.

We add a vertical wind defined as:

$$w(x,y,z,t) = w_0 \cos(\omega T) \tag{24}$$

The vertical wind speed scale is taken as $5\,10^{-2}\,\mathrm{m\,s^{-1}}$, a typical scale for synoptic-scale vertical motion in the troposphere. Since $\frac{\partial u}{\partial x} = \frac{\partial w}{\partial z} = 0$ and since the density field is uniform, this mass flux is non-divergent.

This case can describe a plume that is initially vertical, covering a 50-km wide column (two grid-cells), uniformly spanned over a 3km altitude range (4500 to 7500 m, corresponding to 6 grid cells, see Tab. 2). This initially thick vertical column later evolves under the action of wind shear into a thin layer.

Direct integration of Eq. 24 and then of Eq. 23 give access immediately to the position of a particle initially ($t_i = 0$) at position ($x_i; z_i$):

$$x(t) = x_i + \frac{2U_0}{H} z_i t + \frac{2U_0 W_0}{H \omega^2} (1 - \cos(\omega t)) \tag{25}$$

$$z(t) = z_i + \frac{W_0}{\omega} \sin(\omega t) \tag{26}$$

Eqs. 25-26 describe the superposition of a horizontal motion forced by the horizontal wind speed defined in 23, and an elliptic motion with pulsation $\omega$ due to the oscillating vertical speed (Eq. 24) and its interaction with the horizontal wind shear. The vertical semi-axe of this ellipse is $\frac{w_0}{\omega} \simeq 688\,\mathrm{m}$, and the horizontal semi-axe is $\frac{2U_0 W_0}{H \omega^2} \simeq 9.12 \times 10^3\,\mathrm{m}$.

$(x(t); z(t))$ being, for any given time $t$, affine functions of $(x_i; z_i)$, the wind field of Eqs. 23-24 advects straight lines into straight lines. In particular, the initial rectangular zone containing the tracer will be advected, at any given time, into a parallelogram, which summits are readily given by applying Eqs. 25-26 to the summits of the initial rectangle. Inside this moving parallelogram, that is increasingly tilted with time, tracer mixing ratio is equal to $100\,\mathrm{ppb}$, zero outside, giving access to the exact solution of the case at any time.

### 2.2.2 Case 2: Long-range advection of thin layer

In this case, the initial tracer mixing ratio is as follows:





$$\alpha\left(t=0,x,z\right)=\alpha_m \ \text{ if } -h_2 \le z - \frac{H}{2} \le h_2 \tag{27}$$

$\alpha\left(t=0,x,z\right)=0 \text{ otherwise,} \tag{28}$

with $h_2 = 500\,\text{m}$. These equations describe a zonally infinite layer contained between $z = 5500\,\text{m}$ and $z = 6500\,\text{m}$. As in case 1, $\alpha_m = 100\,\text{ppb}$.

Zonal wind is constant and uniform:

$$u(x,z,t)=U_0=\frac{L}{2T} \tag{29}$$

so that the horizontal motion of the fluid brings it back at its initial $x$ coordinate after time $2T$, which will be the duration of the experiment. Vertical wind is defined as:

$$w(x,y,z,t)=w_0\cos\left(\frac{4\pi x}{L}\right) \tag{30}$$

As for Case 1, $w_0 = 5 \times 10^{-2}\,\text{m s}^{-1}$. This vertical wind speed has two maxima and two minima over the horizontal domain. Integration of Eqs. 29-30 is immediate and give the trajectory of a particle located at $(x_i; z_i)$ at time $t=0$:

$$x\left(t\right)=\frac{Lt}{2T} \tag{31}$$

$$z\left(t\right)=z_i+\frac{W_0 T}{2\pi}\left(\sin\left(\frac{4\pi x_i}{L}+\frac{2\pi t}{T}\right)-\sin\left(\frac{4\pi x_i}{L}\right)\right) \tag{32}$$

In particular, after a time $kT$, k being an integer, all the fluid particles are displaced by distance $\frac{kL}{2}$ in the horizontal direction and back to their initial altitude. At these times, since the initial tracer plume is zonally uniform and infinite, the field of mixing ration will be exactly back to its initial value everywhere.

This case can describe in a simplified way long-range advection of a $1\,\text{km}$ thick layer of an inert tracer under the action of a uniform zonal wind and submitted to the action of alternatively ascendant and subsident winds, representing for example in an extremely simplified way the advection of this layer through synoptic-scale structures. In the atmosphere, such fine layers of tracers are frequently formed by stretching of initially thicker polluted layers, as represented in Case 1.

### 2.2.3   Case 3: Fine layer advection and convergence rate test

This case has been designed to study numerical convergence rate of the various configurations that will be tested as a function of space resolution. The case setup is the same as for Case 2, with wind speeds similar to Eqs. 29-30:

$$u(x,z,t)=U_0=\frac{L}{T} \tag{33}$$





$$w(x,y,z,t) = w_0 \cos\left(\frac{2\pi x}{L}\right) \tag{34}$$

Due to the need of increasing resolution and therefore the numerical cost of simulations, we simulate only one spatial and temporal period of this case (instead of 2 spatial and temporal periods for Case 2), hence the differences between equations 33-34 and 29-30.

The initial tracer mixing ratio is prescribed as:

$$\alpha_i(x,z) = \frac{\alpha_m}{4}\left(1 + \cos\left(\frac{\pi(z - H/2)}{h_3}\right)\right)^2 \text{ if } -h_3 \le z - \frac{H}{2} \le h_3 \tag{35}$$

$$\alpha_i(x,z) = 0 \text{ otherwise,}$$

with $h_3 = 1500\,\mathrm{m}$. This initial mixing ratio distribution has been designed to be $C^2$ (and in fact $C^3$) everywhere. It is therefore smooth enough to permit a convergence experiment with all the transport schemes which rely, at most, on the existence and continuity of the second-order derivative of the transported field (for the PPM scheme).

For convergence rate tests, 5 different resolutions have been tested (Table 2).

### 2.3 Implementation

Implementation of the idealized experiments and transport scheme have been done within the under development ToyCTM code. ToyCTM is a Python code targeting chemistry-transport studies in academic cases. So far, ToyCTM relies on classical numpy arrays. Its Object-Oriented design provides a class structure enabling extensibility, *i.e.* users can easily code new transport schemes or define personal grid geometry. A basic chemistry module is present and allows one to test chemical reactions on top of transport. See the **Code availability** paragraph at the end of the manuscript for acess to the code version used for the present study and to the current development version of the code.

## 3 Results

### 3.1 Case 1: Thin layer formation under wind shear

Figure 1 shows the outputs of the four simulations realized with different schemes for vertical advection. All four simulations succeed in reproducing the tilted orientation of the final plume and its location, but differ greatly in their maximal value and spatial extension, with the smallest maximal value of mixing ratio for the upwind scheme, and the maximal value for the DL99 scheme, the Van Leer and PPM schemes ranging in-between. From a quantitative point of view (Tab. 3, the Upwind, VL and PPM schemes perform as could be expected from their order of accuracy, with the third-order PPM scheme performing better than the second-order Van Leer scheme and the first-order Upwind scheme in terms of all the diagnostics that have been calculated. More surprisingly, the first-order DL99 scheme performs better than all these schemes in terms of all these



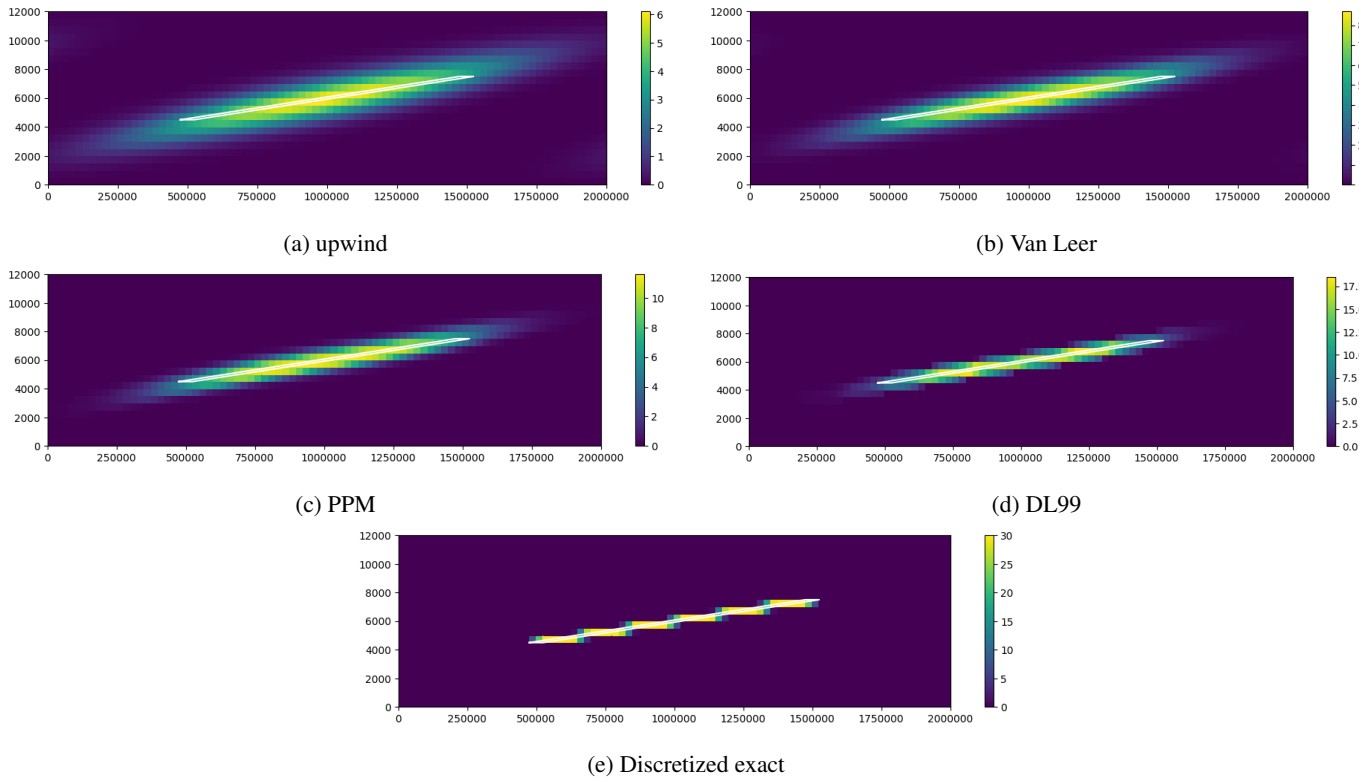

**Figure 1.** Final state of the numerical simulation for case 1 after simulations. Panels (a), (b), (c), (d) show the results obtained with Upwind, VL, PPM and DL99, respectively, panel (e) represent the exact solution discretized on model grid. On panels (a), (b), (c), (d), (e), the contour of the exact solution is materialized by a white parallelogram.

|  | Exact | Upwind | VL | PPM | DL99 |
|---|---|---|---|---|---|
| Max. MR | 30.0 | 6.10 | 8.69 | 11.6 | 18.5 |
| % error (norm 1) | 0. | 157. | 140. | 122. | 87. |
| % error (norm 2) | 0. | 86.1 | 80.4 | 73.9 | 60.3 |
| % mass in envelope | 100.0 | 23.3 | 33.2 | 44.4 | 64.7 |

**Table 3.** Performance of simulations performed with the Upwind, VL, PPM and DL99 vertical advection schemes relative to the discretized exact solution for Case 1: percent relative error in $\| \cdot \|_1$ and $\| \cdot \|_2$ and percent of total tracer mass contained in the correct envelope

diagnostics, by a wide margin: in this case study, the performance gain of DL99 relative to PPM is similar to the gain of PPM relative to Godunov in terms of maximal mixing ratio, percentage of mass in the evvelope and accuracy.



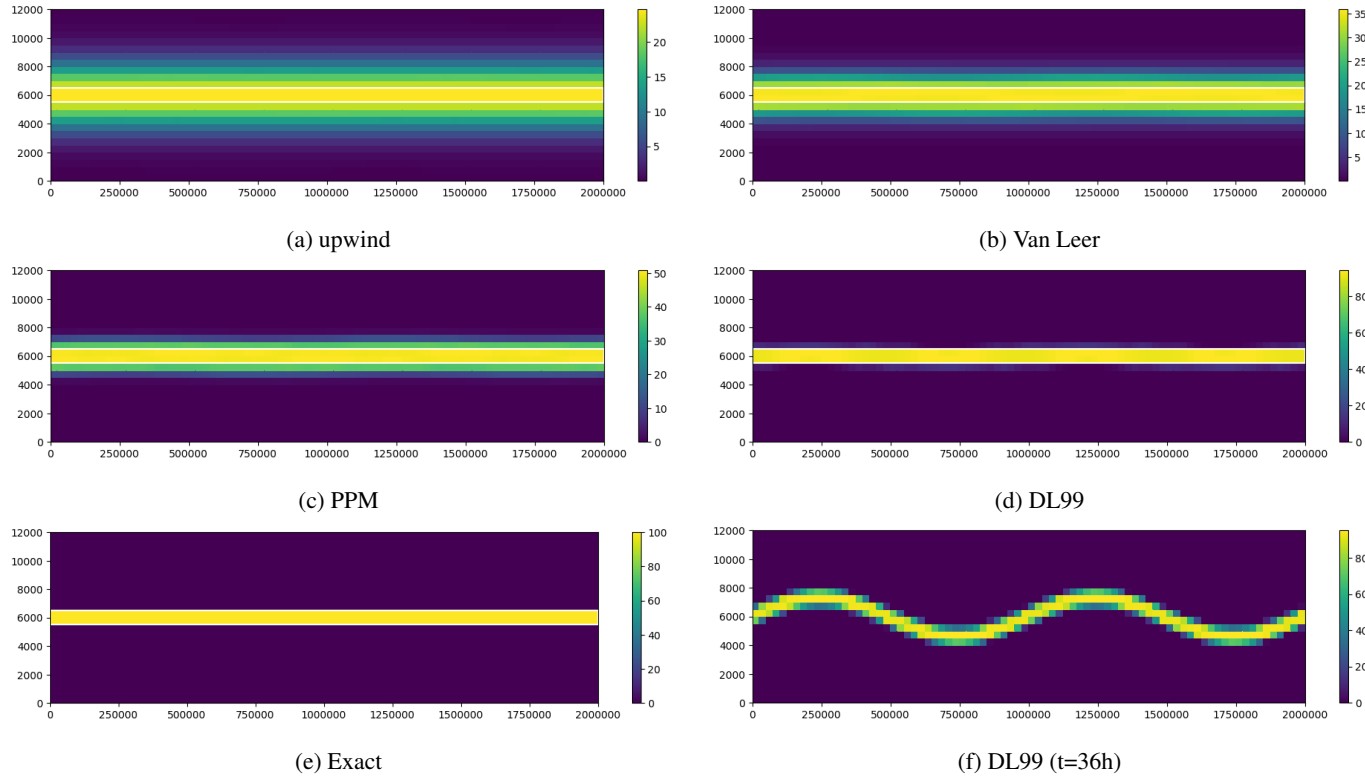

| (a) upwind | (b) Van Leer |
| (c) PPM | (d) DL99 |
| (e) Exact | (f) DL99 (t=36h) |

**Figure 2.** Final state of the numerical simulation for case 2 after simulations. Panels (a), (b), (c), (d) show the results obtained with Upwind, VL, PPM and DL99, respectively, panel (e) represents the exact solution (which is strictly equal to the initial state due to periodicity in time of the case), and panel (f) represents the state of the DL99 simulation after 36h simulation.

### 3.2 Case 2: Long-range advection of thin layer

Figure 2 shows the final state of Case 2 simulation for the four advection schemes that have been tested, as compared to the exact solution. Visually, the Després and Lagoutière (1999) scheme has performed best in bringing virtually all the tracer back
into its original envelope after two complete vertical oscillations. Its performance in this case is best of all the tested schemes (Table 4), with only a 6% reduction in the maximal value of tracer mixing ratio (49% with the PPM method, even more with the Godunov and Van Leer schemes), and very small error values in $\| \cdot \|_1$ and $\| \cdot \|_2$ compared to the other tested schemes. 93% of the mass is contained in the theoretical envelope where it should be after the end of the numerical experiment (50% only with the PPM scheme).

### 3.3 Case 3: Fine layer advection and convergence rate test

The numerical convergence rates of all the tested advection configurations for $\| \cdot \|_1$ and $\| \cdot \|_2$ based on the last segment in Figs. 3a-b (between $n_x = 160$ and $n_x = 320$) are shown in Tab. 5. In $\| \cdot \|_1$. These orders of convergence are around 0.8 for





|  | Exact | Upwind | VL | PPM | DL99 |
|---|---|---|---|---|---|
| Max. MR | 100. | 24.7 | 35.9 | 50.8 | 94.2 |
| % error (norm 1) | 0. | 151. | 129. | 99.4 | 14.4 |
| % error (norm 2) | 0. | 82.6 | 74.6 | 63.3 | 11.2 |
| % mass in envelope | 100. | 24.7 | 35.6 | 50.3 | 92.8 |

**Table 4.** Performance of simulations performed with the Upwind, VL, PPM and DL99 vertical advection schemes relative to the exact solution for Case 2: percent relative error in $\|\cdot\|_1$ and $\|\cdot\|_2$ and percent of total tracer mass contained in the correct envelope

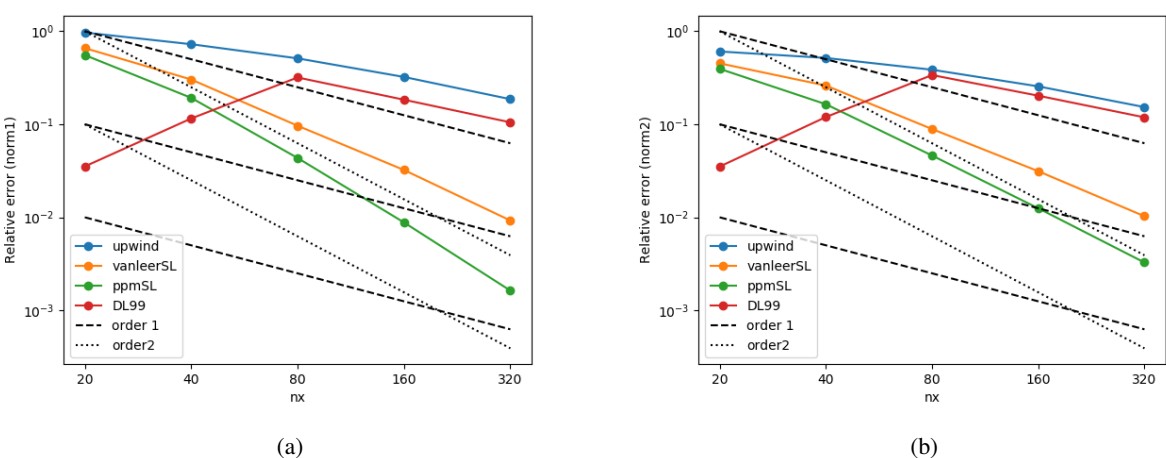

(a)                                                           (b)

**Figure 3.** Convergence rate results for the four tested vertical advection schemes as a function of the number of horizontal points $n_x$ in the horizontal direction, in $\|\cdot\|_1$ (panel a) and $\|\cdot\|_2$ (panel b) for Case 3. The black dashed lines shox the slope expected for order-1 (long-dash) and order-2 (short-dash)

the DL99 and Upwind schemes because vertical resolution is still insufficient even at this high resolution to ensure theoretical convergence for these schemes. The Van Leer scheme yields a convergence rate of 1.80 in $\|\cdot\|_1$, 2.43 for the PPM scheme. A
smoother test case has been performed to check that all schemes are able to obtain convergence up to their theoretical order (see Appendix).

Figure 3 shows that, when resolution is too coarse to appropriately resolve the thin layer, the Després and Lagoutière (1999) scheme strongly overperforms higher-order schemes and offers an accuracy that is substantially better than both the Van Leer (1977) scheme and the Colella and Woodward (1984) scheme. This is consistent with the results obtained on Cases 1 and 2,
and explains why in these cases the Després and Lagoutière (1999) scheme permits to obtain excellent results in reproducing the thin layer of tracer. On the other hand, when vertical resolution becomes sufficient to appropriately resolve the smooth tracer layer, higher-order schemes perform better than the Després and Lagoutière (1999) schemes, which in turns falls back towards an accuracy similar to the Godunov scheme, consistent with its expected order or accuracy.





| Horiz. advection scheme | Vert. advection scheme | $\|\cdot\|_1$ convergence rate | $\|\cdot\|_2$ convergence rate |
|---|---|---|---|
| PPM | Upwind | 0.79 | 0.74 |
| PPM | DL99 | 0.81 | 0.77 |
| PPM | Van Leer | 1.80 | 1.60 |
| PPM | PPM | 2.43 | 1.94 |

**Table 5.** Convergence rates in $\|\cdot\|_1$ and $\|\cdot\|_2$

## 4 Discussion

The numerical experiments that have been presented confirm the interest of using the Després and Lagoutière (1999) scheme for vertical transport in chemistry-transport models, as already claimed by Lachatre et al. (2020). The idealized framework set up here permits to examine some of the questions that were out of reach in the real case of Lachatre et al. (2020) due to uncertainties on the forcing meteorological fields, the volcanic emissions and to the lack of accurate measurement of plume structure.

First, we show not only that the Després and Lagoutière (1999) scheme is less diffusive than Van Leer (1977) and Colella and Woodward (1984), as could be expected due to its antidiffusive design, but also that, in the presence of sharp vertical gradients that are not adequately resolved at model resolution, using this vertical transport scheme also increases model accuracy compared to the exact solution. This improvement is spectacular for both Case 1 (Tab. 3 and Fig. 1) representing formation of a thin polluted layer under the action of wind shear and Case 2 (Tab. 4 and Fig. 2) representing long-range advection of

a thin polluted layer. For both these cases, the objective scores as well as the visual comparison of the simulated final state with the exact solution show that, at this resolution and for these cases, using Després and Lagoutière (1999) both reduces diffusion and increases accuracy compared to the schemes of Godunov, Van Leer (1977) and Colella and Woodward (1984). While reduction of diffusion is in line with the results of Lachatre et al. (2020) and could be expected by design of the Després and Lagoutière (1999) schemes, improved accuracy in presence of sharp gradients is a strong argument in favor of using the

Després and Lagoutière (1999) scheme for vertical transport in chemistry-transport models.

Improved accuracy of a low-order scheme compared to higher-order schemes for a given resolution is not impossible from a theoretical point of view but still counterintuitive since higher-order schemes are expected to reduce numerical error at any given resolution compared to lower-order schemes due to "smarter" reconstruction procedures. To understand this surprisingly good behaviour of Després and Lagoutière (1999) compared to teh higher-order Van Leer (1977) and Colella and Woodward

(1984) schemes, we have performed a convergence test for advection of a 3000m-thick layer with a smooth vertical profile. This convergence test (Fig. 3) show that the Després and Lagoutière (1999) performs better than these classical order-2 schemes if model vertical resolution $\Delta z$ is equal to $1000\,m$ or $500\,m$, but that due to their faster convergence rate, order-2 schemes performs better if $\Delta z \leq 250\,m$. In more general words, this result suggest that the Després and Lagoutière (1999) scheme may be expected to perform better than classical schemes in chemistry-transport models for advection of polluted plume thinner



than $\simeq 6\,\Delta z$, while higher-order schemes can be expected to perform better for advection of polluted plumes thicker than $6\,\Delta z$. When vertical resolution becomes much too fine compared to the size of the modelled object (polluted plume thicker that $\simeq 50\,\Delta z$), accuracy of the simulations with the Després and Lagoutière (1999) scheme stops improving with resolution at order-1 rate (Fig. A1). Examination of the simulation outputs for these configuration reveal that progress in accuracy is hindered by the apparition of undesirable small-scale oscillations that degrade acccuracy (not shown).

**5  Conclusions**

The Després and Lagoutière (1999) scheme is shown to offer excellent performance in reproducing long-range advection of fine tracer layers when vertical resolution is so coarse that a correct representation of these layers could be expected to be impossible in practice. This improved performance of Després and Lagoutière (1999) compared to Van Leer (1977) and Colella and Woodward (1984) translates into reduced numerical diffusion and in improved accuracy compared to these higher-order

schemes. Convergence tests show that this improved performance exists only for tracer layers that are represented by less than 6 grid cells in the vertical direction, while for finer resolutions higher-order schemes perform better due to their faster convergence towards the exact solution. In other terms, if model resolution is fine enough to represent properly the plume, then higher-order schemes are still a better choice.

We think that these results are important because they explain under which conditions the Després and Lagoutière (1999)

is able to reduce excessive vertical diffusion in CTMs, as was observed in Lachatre et al. (2020), show that this reduction in numerical diffusion is not obtained at the expanse of accuracy, which is improved as well. Since the vertical resolution of chemistry-transport model is usually coarse in the free troposphere, while the advected plumes tend to be extremely thin in this part of the atmosphere due to the action of wind shear, the present study along with Lachatre et al. (2020) advocates for using the Després and Lagoutière (1999) transport scheme for chemistry-transport modelling in the free troposphere, and probably

even more in the stratosphere where vertical diffusion needs to be extremely reduced. More investigation is needed in real and/or idealized cases to adress several questions:

- Does the Després and Lagoutière (1999) perform better in the boundary layer, where CTM resolution is typically finer than in the free troposphere ?

- If not, is it possible to use a traditional transport scheme in the boundary layer and the Després and Lagoutière (1999)

scheme in the free troposphere without intruducing numerical artefacts in the buffer zone ?

- Atmospheric chemistry being a non-linear process, how does reduction of excessive numerical diffusion in the troposphere affect representation of chemistry inside chemically active air masses such as volcanic or biomass burning plumes.

- Is it desirable, and under which conditions, to use antiffusive transport schemes in the horizontal directions as well ?





*Code availability.* ToyCTM is free software distributed under the GNU General Public License v2. The exact version used for the present study (Mailler and Pennel, 2020) is available permanently in the HAL repository at https://hal.archives-ouvertes.fr/hal-02933095. The latest stable version of toyCTM is available from: https://gitlab.in2p3.fr/ipsl/lmd/intro/toyctm/-/archive/master/toyctm-master.tar.gz.



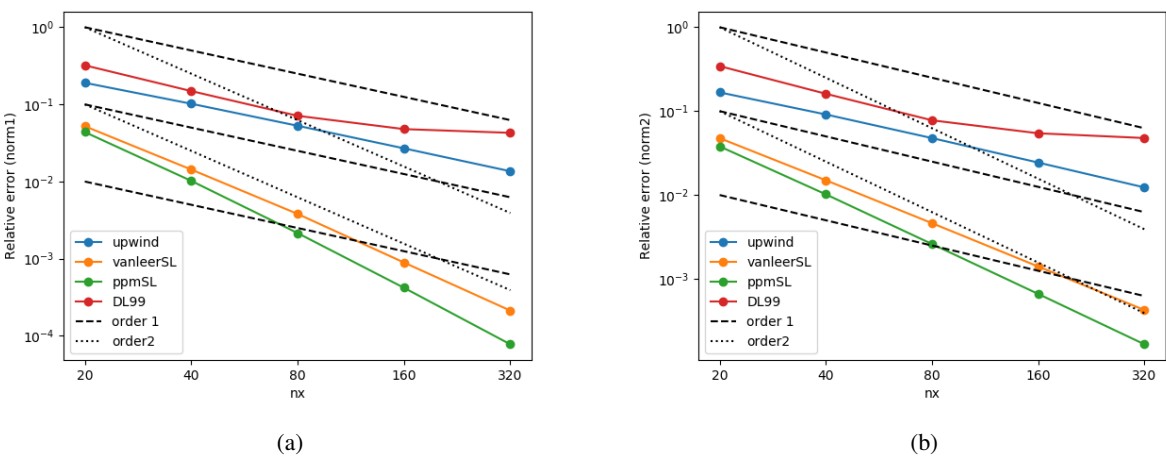

(a)                                (b)

**Figure A1.** Convergence rate results for the four tested vertical advection schemes as a function of the number of horizontal points $n_x$ in the horizontal direction, in $\|\cdot\|_1$ (panel a) and $\|\cdot\|_2$ (panel b) for Case 3

## Appendix A: Case 4: Convergence test in a smooth case

### A1 Case definition

This case has been designed to study numerical convergence rate of the various configurations that will be tested as a function of space resolution. The case setup is the same as for Cases 2 and 3, with wind speeds from Eqs. 29-30, but the initial tracer mixing ratio is prescribed as:

$$\alpha_i(x,z) = \frac{\alpha_m}{4}\left(1+\cos\left(\frac{\pi(z-H/2)}{h_4}\right)\right)\left(1+\cos\left(\frac{\pi(x-L/2)}{\delta}\right)\right) \tag{A1}$$

with $h_4 = \frac{H}{2} = 6000\,\mathrm{m}$ and $\delta = \frac{L}{2} = 5\times10^5\,\mathrm{m}$. This initial tracer distribution represents a cosine bell centered at $\left(x=\frac{L}{2}; z=\frac{H}{2}\right)$,
$C^\infty$ everywhere with extremely smooth variations in space. Domain resolution and sizes are as shown in Tab. 2.

### A2 Results

Table A1 and Fig. A1 show that all the tested configurations exhibit the expected convergence order at least in $\|\cdot\|_1$, but accuracy with the DL99 scheme stops improving when vertical resolution becomes "too fine" compared to the thickness of the represented layer. In the present case, this "saturation" of convergence occurs between $nx = 80$ $(nz = 48)$ and $nx = 160$
(nz=96):

*Author contributions.* All the authors have contributed to the design of the simulated cases, SM has performed and analyzed the simulations, SM has developed the software with RP, LM, ML and RP have contributed to writing and improving the manuscript.





| Horiz. advection scheme | Vert. advection scheme | $\|\cdot\|_1$ convergence rate | $\|\cdot\|_2$ convergence rate |
|---|---|---|---|
| PPM | Upwind | 0.99 | 0.98 |
| PPM | DL99 | 1.06* | 1.05* |
| PPM | Van Leer | 2.07 | 1.72 |
| PPM | PPM | 2.43 | 1.99 |

**Table A1.** Convergence rates in $\|\cdot\|_1$ and $\|\cdot\|_2$ for Case 4. Convergence rates for the DL99 scheme, marked with a * symbol, are evaluated between $nx = 40$ and $nx = 80$, before numerical errors appear and begin to degrade the result.

*Competing interests.* None.

*Acknowledgements.* This study has been supported by AID (Agence de l'Innovation de Défense) under grant TROMPET. The simulations
have been performed at the ESPRI/IPSL data center and at TGCC under GENCI A0070110274 allocation.





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
