# Peer review of "Using the Després and Lagoutière (1999) antidiffusive transport scheme: a promising and novel method against excessive vertical diffusion in chemistry-transport models."

_Geoscientific Model Development, 2020_

## Referee Comment (RC1) · Anonymous Referee #1 · 3 Dec 2020

The authors use idealized tests to quantify the performance of the Després and Lagoutière (DL99) advection scheme for the calculation of vertical constituent transport in chemistry-transport settings, comparing it to both classical schemes such as Van Leer and (relatively) modern schemes such as the Piecewise Parabolic Method (PPM). They find that DL99 may be able to produce a more accurate simulation of the transport of thin atmospheric layers than the standard PPM or Van Leer approaches, in spite of being formally lower-order in accuracy. This is an interesting and potentially highly significant result, due to the known difficulties of simulating such layers in the atmosphere and the computational cost of increasing the vertical model resolution – the only serious solution yet suggested elsewhere in the literature.

The central question addressed by the authors, of how to efficiently address the issue of numerical diffusion in CTMs, is important and timely. Their use of an idealized test to complement their earlier tests in a "real-world" situation is appropriate, both in scientific terms and for GMD. Their conclusions are well supported by the data, with the exception of one comment (see below). The identification that an efficient, first-order accurate vertical advection scheme might be able to help address the long-standing issue of numerical diffusion is a significant advancement in the field.

Overall, I found this manuscript to be significant, well-written, and concise. I therefore recommend it for publication, once a small number of issues are addressed. I have sorted my comments into major, minor, and superficial.

Major (substantive) comments:

1. While I understand that the Upwind and DL99 schemes are first-order, it seems like an unnecessary confounding to use different operator splitting methods for these two methods in comparison to the VL and PPM schemes. It would be useful to see a comparison where all schemes are using (e.g.) the Strang operator splitting. Although this is not expected to yield an improvement in the Upwind and DL99 results, it would at least verify that the improved performance is not because of the operator splitting approach. Given the performance characteristics of DL99 (i.e. low sensitivity to small CFL numbers) one would hope that Strang's scheme also would not compromise its accuracy, although it might incur an unnecessary expense.

2. It seems like an oversight to not invoke Godunov's theorem (Godunov, 1959), especially on lines 306-308. It is a known result that any higher-order scheme cannot exceed first-order accuracy in the vicinity of a sharp gradient, so it is not true that "higher-order schemes are expected to reduce numerical error at any given resolution".

3. On line 327, the authors state that "if model resolution is fine enough to represent properly the plume, then higher-order schemes are still a better choice", but I am not

sure this is true (or that this manuscript even supports that conclusion). An important point they raised is that the DL99 scheme does a good job even in the situation of low CFL numbers (line 140), and it seems that such conditions are likely to be common when considering vertical movement in the atmosphere. I would recommend that this conclusion be removed or at least made more precise to account for the fact that it may only be true under certain conditions. This is hinted at through the final sentence of the discussion (line 318) but the authors are understating the importance of this point. The implication that increasing vertical resolution may be an inefficient solution for even higher-order methods is a potentially significant finding.

I also have some minor comments:

1. Figures 1 and 2 would be improved by using the same color scale for all panels (i.e. 0-20 ppb for Figure 1, and 0-100 ppb for Figure 2)

2. I would suggest that the authors consider replacing "promising novelty" in the title with, say, "promising and novel solution to ". I think that "novelty" makes the work sound unimportant, whereas I found this work to be intriguing and of high value.

Finally, I tried to make a note of any typos or grammatical errors I found. However, I would suggest that the authors make an additional sweep for grammatical accuracy:

1. Line 9: "an important direction into improvement" doesn't quite make sense. Perhaps "necessary step in the development"? 2. Line 17: "too much observations" should be "too much compared to observations" 3. Line 72: "permit" should be "ensure" or similar 4. Line 73: there is a spurious space between the closing bracket and comma. 5. Figure 3 caption: "shox" should be "show" 6. Line 309: "teh" should be "the" 7. Line 328: "enaugh" should be "enough" 8. Line 336: "adress" should be "address" 9. Throughout: "1d" should be "1D" or "1-D" 10. Throughout: some language is somewhat nonscientific (e.g. "spectacular" on line 298 is hyperbolic)

References

Godunov, S. K.: A difference method for numerical calculation of discontinuous solutions of the equations of hydrodynamics, Matematicheskii Sbornik, 47(3), 271–306, 1959.

---

## Referee Comment (RC2) · Anonymous Referee #2 · 21 Dec 2020

[referee-annotated manuscript omitted]

---

## Short Comment (SC1) · 21 Dec 2020

Dear authors,

in my role as Executive editor of GMD, I would like to bring to your attention our Editorial version 1.2:

https://www.geosci-model-dev.net/12/2215/2019/

This highlights some requirements of papers published in GMD, which is also available on the GMD website in the 'Manuscript Types' section:

http://www.geoscientific-model-development.net/submission/manuscript_types.html

[Figure]

In particular, please note that for your paper, the following requirements have not been met in the Discussions paper:

- "The main paper must give the model name and version number (or other unique identifier) in the title."

In order to simplify reference to the "antidiffusive transport scheme", please add a name or acronym and a version number in the title of your article in your revised submission to GMD.

Yours,

Astrid Kerkweg

---

## Author Comment (AC2) · 12 Feb 2021

**Answer to Referee #2**

Sylvain Mailler[1,2], Romain Pennel[1], Laurent Menut[1], and Mathieu Lachâtre[1]

[1]*LMD/IPSL, École Polytechnique, Institut Polytechnique de Paris, ENS, PSL Research University, Sorbonne Université, CNRS, Palaiseau France*
[2]*École des Ponts-ParisTech, Marne-la-Vallée, France*

Referee #2 has brought our attention onto several points in our formulations and typographic setup. We are thankful for the attention he brought to these details of the work, and have corrected the points he has raised accordingly.

**spectacular $->$ important ?**
As also noted by Reviewer 1, words as "spectacular" are rather non-scientific. This sentence has been reworded as:
The increase of accuracy and the reduction of diffusion are substantial when . . .

**suggest, describes : plural / singular ?**
As noted our use of plural / siingular for publications with more that one author was inconsistent. The correct use is plural (e.g. Lachatre et al. (2020) describe). This is corrected.

**No extra space after equations**
This has been corrected (except in places in which a paragraph ended, materialized by a dot (.) after the equation. As also highlighted by the reviewer, dots (.) were missing after some equations that end a sentence / paragraph. We have added them where needed throughout. We believe that the Editorial office can help us further in correcting the typographic layout of equations and their environment, shall that be needed.

**No extra space before comma (p. 4)**
This has been corrected.

**l. 126, extra parenthesis**
this has been removed.

**l. 131, missing dot**
Corrected.

**l. 194, $w_0 -> W_0$**
Corrected.

**Fig. 1 : should be constant $z$ ?? Check (26)**

As described in lines 207-211 of the manuscript, Eqs. 25-26 describe how the initially rectangular zone containing the tracer is transformed into a tilted parallelogram under the action of wind shear. The figure is in line with Eqs. (25)-(26) and the description in lines 207-211

**References**

Lachatre, M., Mailler, S., Menut, L., Turquety, S., Sellitto, P., Guermazi, H., Salerno, G., Caltabiano, T., and Carboni, E.: New strategies for vertical transport in chemistry-transport models: application to the case of the Mount Etna eruption on March 18, 2012, Geophys. Model Dev. Discuss., 2020.

---

## Author Comment (AC3) · 12 Feb 2021

Dear Astrid Kerkweg,

Thank you for reminding us for the requirements of *Geosci Model Dev*, including two shortcomings in our title:

1. "The main paper must give the model name and version number (or other unique identifier) in the title." : Here we feel that the model (toyCTM) is just a *ad hoc* framework that we have built in order to test some changes in transport strategy. The choice to use a *ad hoc* framework for this study is a way for us to make clear that the application of

the methodology we propose here is actually independant of any particular CTM. This corresponds to the appreciation of Referee 1 : "The central question addressed by the authors, of how to efficiently address the issue of numerical diffusion in CTMs, is important and timely". Therefore, we would largely prefer not to make the reader feel in any way that the transport strategy we propose is linked to this particular framework. Actually, we feel that our findings have the potential for a wide use in chemistry-transport models, not at all restricted to the present academic case (this feeling seems to be shared by Referee 1 as well).

2. "In order to simplify reference to the "antidiffusive transport scheme", please add a name or acronym and a version number in the title of your article in your revised submission to GMD." : To take this comment into account as well as a suggestion of Referee 1 to rephrase the title, we propose to resubmit the manuscript with the following updated title, including the exact reference to the transport scheme we test :

"Using the Després and Lagoutière (1999) antidiffusive transport scheme: a promising and novel method against excessive vertical diffusion in chemistry-transport models.". This scheme has no widely recognised name so far, so that we feel that the reference to its original authors is the best accurate reference to it.

---

## Author Response (AR1)

**Anthor's response concerning manuscript formerly titled "Using an antidiffusive transport scheme in the vertical direction : a promising novelty for chemistry-transport models"**

Sylvain Mailler[1,2], Romain Pennel[1], Laurent Menut[1], and Mathieu Lachâtre[1]

[1] *LMD/IPSL, École Polytechnique, Institut Polytechnique de Paris, ENS, PSL Research University, Sorbonne Université, CNRS, Palaiseau France*
[2] *École des Ponts-ParisTech, Marne-la-Vallée, France*

**Contents**

We are very grateful of the thorough review performed by the Referees and we would like to thank them for their very encouraging opinion on our work. Due to the useful suggestion of Referee #1 of using the same splitting strategy for all numerical experiments to increase confidence in our results, we have redone most of the calculations that were presented. The resulting differences, however, are very small, and do not change the interpretation, strengthening our confidence in the results we present. We are also grateful to the Editor for the useful reminder of GMD policies, helping us propose a clearer and more accurate title for the revised version. We have performed all the changes that were asked

by the Reviewers and the Editor, and believe that the review process has greatly helped us improve our work. We hope that this new, improved version of our manuscript will fit the standards for publication in *Geosci. Model Dev.*. In the

rest of this document, contributions of the Referees and the Editor are in **bold font**, and the changes brought to the manuscript are highlighted in blue.

**1    Improved title**

Comments by the Editor and by Referee #1 led us to change the manuscript title. Referee #1 suggested the following change: "**I would suggest that the authors consider replacing "promising novelty" in the title with, say, "promising and novel solution to". I think that "novelty" makes the work sound unimportant, whereas I found this work to be intriguing and of high value**". Editor requested that, in conformity with GMD policy, we should "**in order to simplify the reference to the "antidiffusive transport scheme", [please] add a name or acronym and a version number in the title of your article in your revised submission to GMD**".

Following these two requests, the title of the manuscript has been changed from "Using an antidiffusive transport scheme in the vertical direction : a promising novelty for chemistry-transport models" to "Using the Després and Lagoutière (1999) antidiffusive transport scheme: a promising and novel method against excessive vertical diffusion in chemistry-transport models.".

**2 Answers to Referee #1**

**2.1 Answers to Major comments**

> **1. While I understand that the Upwind and DL99 schemes are first-order, it seems like an unnecessary confounding to use different operator splitting methods for these two methods in comparison to the VL and PPM schemes. It would be useful to see a comparison where all schemes are using (e.g.) the Strang operator splitting. Although this is not expected to yield an improvement in the Upwind and DL99 results, it would at least verify that the improved performance is not because of the operator splitting approach. Given the performance characteristics of DL99 (i.e. low sensitivity to small CFL numbers) one would hope that Strangs scheme also would not compromise its accuracy, although it might incur an unnecessary expense.**

We had a discussion among authors on this point before initial submission, and decided to use each scheme "at its best", in the configuration that would be used naturally for a simulation. So, the first-order schemes with a Lie splitting not to incur into excess computational time and not to degrade performance by useless splitting of horizontal time steps, and the second-order schemes with a Strang splitting to preserve their accuracy.

Even though we have been careful to split the horizontal time stepping (which is performed with PPM in all configurations) and not the vertical one, precisely not to lose the ability to compare the runs with each other, we agree with the Referee that this choice adds some uncertainty and may cast doubt on the comparisons we present. Therefore, as the Referee suggests, we have redone the calculations using Strang splitting for all schemes, maintaining the same choice to always split horizontal integration rather than vertical integration. All numbers and plots are changed accordingly in the revised version for the Upwind and DL99 simulations[1]. Below, we introduce and comment the modifications in Tables 3 and 4 brought by this change in the splitting strategy.

Tables 3 and 4 show that a small part of the improvement obtained by using the Després and Lagoutière (1999) scheme instead of the Colella and Woodward (1984) PPM scheme was indeed due to the different splitting approach, as the Referee suggests. This can interpreted as a small amount of additional horizontal diffusion due to splitting horizontal integration. While this additional horizontal diffusion is marginal compared to the strong vertical diffusion in the upwind simulation, it does bring a degradation of a few percents in the performance of the DL99 configutaion, for both simulated cases relative to the Lie splitting.
* * *
[1]The numbers for the VL and PPM runs in tables 3-4 were affected by an error (probably due to making these calculations with an earlier model version). The differences presented in tables 3 and 4 of the present document are relative to the corrected values.

|  | Exact | Upwind | VL | PPM | DL99 |
|---|---|---|---|---|---|
| Max. MR | 30.0 |  **6.11** | 10.3 | 11.7 |  **18.2** |
| % error (norm 1) | 0. | 157. | 131. | 122. |  **87.6** |
| % error (norm 2) | 0. | 86.1 | 76.9 | 73.8 |  **60.4** |
| % mass in envelope | 100.0 | 23.3 | 39.0 | 44.6 |  **64.9** |

Table 3: Performance of simulations performed with the Upwind, VL, PPM and DL99 vertical advection schemes relative to the discretized exact solution for Case 1: percent relative error in $\|\cdot\|_1$ and $\|\cdot\|_2$ and percent of total tracer mass contained in the correct envelope. The numbers that are not changed up to the third-figure truncature appear in normal fonts, the numbers that are changed appear in bold font, and the former value appears in striked-out font.

|  | Exact | Upwind | VL | PPM | DL99 |
|---|---|---|---|---|---|
| Max. MR | 100. | 24.7 | 42.2 | 50.8 |  **92.6** |
| % error (norm 1) | 0. | 151. | 116. | 99.3 |  **18.8** |
| % error (norm 2) | 0. | 82.6 | 69.8 | 63.2 |  **14.2** |
| % mass in envelope | 100. | 24.7 | 42.0 | 50.3 |  **90.6** |

Table 4: Performance of simulations performed with the Upwind, VL, PPM and DL99 vertical advection schemes relative to the exact solution for Case 2: percent relative error in $\|\cdot\|_1$ and $\|\cdot\|_2$ and percent of total tracer mass contained in the correct envelope. The numbers that are not changed up to the third-figure truncature appear in normal fonts, the numbers that are changed appear in bold font, and the former value appears in striked-out font.

This degradation is however very small compared to the difference between DL99 ans, e.g., PPM, so that the conclusions of the study are not changed.

> **2. It seems like an oversight to not invoke Godunov's theorem (Godunov and Bohachevsky, 1959), especially on lines 306-308. It is a known result that any higher-order scheme cannot exceed first-order accuracy in the vicinity of a sharp gradient, so it is not true that "higher-order schemes are expected to reduce numerical error at any given resolution".**

We are grateful to the Reviewer for drawing our attention to the need for citing the seminal paper of Godunov and Bohachevsky (1959). The result most widely known as Godunov's theorem states that a linear, monotonous scheme, cannot exceed first-order convergence in accuracy. This is why, to ensure monotonicity, higher-order schemes such as Van Leer (1977) or Colella and Woodward (1984), among many others, have to include non-linear "slope limiters" in order to ensure monotonicity, which breaks their linearity.

We have introduced a discussion of our results in light of Godunov's results

in the Discussion section. This discussion reads as follows:

Theory imposes that, when accuracy becomes fine enough, and if the tracer field is smooth, higher-order schemes perform better than lower-order schemes. However, as shown by Godunov and Bohachevsky (1959), linear higher-order schemes cannot be monotonous, a property usually known as Godunov's theorem. This is why, to ensure monotonicity, the schemes of Van Leer (1977) and Colella and Woodward (1984) include non-linear slope-limiters which are activated in the vicinity of extrema and discontinuities. In the vicinity of discontinuities, these formulations introduce large inaccuracies: in these schemes, the use of slope-limiters introduce large errors in the vicinity of discontinuities, and these errors generate excessive numerical diffusion, which is visible in Figs. 1 and 2. On the other hand, as discussed by its creators, the Després and Lagoutière (1999) scheme is designed to reduce numerical diffusion in these areas of steep gradients, which explains why it performs better than Van Leer (1977) and Colella and Woodward (1984) in all respects for cases 1 and 2, which describe discontinuous tracer layers (Tables 3 and 4).

> **3. On line 327, the authors state that "if model resolution is fine enough to represent properly the plume, then higher-order schemes are still a better choice", but I am not sure this is true (or that this manuscript even supports that conclusion). An important point they raised is that the DL99 scheme does a good job even in the situation of low CFL numbers (line 140), and it seems that such conditions are likely to be common when considering vertical movement in the atmosphere. I would recommend that this conclusion be removed or at least made more precise to account for the fact that it may only be true under certain conditions. This is hinted at through the final sentence of the discussion (line 318) but the authors are understating the importance of this point. The implication that increasing vertical resolution may be an inefficient solution for even higher-order methods is a potentially significant finding.**

The missing point in the lines that are cited is that the statements such as "if model resolution is fine enough to represent properly the plume, then higher-order schemes are still a better choice" are true if the underlying tracer field is smooth. In this case (and only in this case), theory guarantees that if the resolution is fine enough, then error becomes smaller for higher-order schemes. In presence of shocks in the tracer concentration and/or its derivatives, such statements are false.

The statement cited by the Referee has been precised: "Theory imposes that, if model resolution is fine enough and if the tracer field is smooth, higher-order

schemes should be more accurate than lower-order schemes.".

A more thorough discussion has also been added to take into account the other reflexions of the Referee:

"In more general words, this result suggest that the Després and Lagoutière (1999) scheme may be expected to perform better than classical schemes in chemistry-transport models for advection of polluted plume thinner than $\simeq 6\,\Delta z$ ($\Delta z$ being the model's vertical resolution), while higher-order schemes can be expected to perform better for advection of polluted plumes thicker than $6\,\Delta z$ if we suppose that the plume has a smooth vertical profile. Under realistic conditions of wind shear, these conditions of sufficient smoothness and thickness might actually be very difficult to reach since, as described in Case 1, vertical wind shear tend to the permanent thinning of atmospheric plumes (this question is discussed in detail in Zhuang et al. (2018)) so that the Després and Lagoutière (1999) may frequently overperform classical order-2 schemes in realistic wind conditions including wind shear."

As suggested, the good behaviour of the Després and Lagoutière (1999) scheme at low CFL numbers is now highlited in a stronger way in the Conclusion: "It is also worth noting that Després and Lagoutière (1999) have shown that their scheme maintains its convergence and low-diffusion properties even if the CFL number becomes small, which is very common for vertical advection in the free troposphere due to typically small vertical speed of air motion (typically a few $\mathrm{cm\,s^{-1}}$)."

**2.2 Answers to Minor comments**

**1. Figures 1 and 2 would be improved by using the same color scale for all panels (i.e. 0-20 ppb for Figure 1, and 0-100 ppb for Figure 2)**

The colorscales have been changed as the Reviewer suggests (improving readability). See Figs. 1-2 in the revised manuscript.

**2. I would suggest that the authors consider replacing promising novelty in the title with, say, promising and novel solution to . I think that novelty makes the work sound unimportant, whereas I found this work to be intriguing and of high value.**

The modification has been done accordingly, see Section 1. The new title is now:

"Using the Després and Lagoutière (1999) antidiffusive transport scheme: a promising and novel method against excessive vertical diffusion in chemistry-transport models."

**2.3  Typos**

Finally, I tried to make a note of any typos or grammatical errors I found. However, I would suggest that the authors make an additional sweep for grammatical accuracy:

1. Line 9:"an important direction into improvemen" doesn't quite make sense. Perhaps "necessary step in the development"?

Changed accordingly.

2. Line 17: "too much observations" should be "too much compared to observations"

Changed accordingly.

3. Line 72: "permit" should be "ensure" or similar

Changed accordingly.

4. Line 73: there is a spurious space between the closing bracket and comma.

Space has been removed.

5. Figure 3 caption: "shox" should be "show"

Changed accordingly

6. Line 309: "teh" should be "the"

Changed accordingly (line 309 and in another occurence)

7. Line 328: "enaugh" should be "enough"

Changed accordingly

8. Line 336: "adress" should be "address"

Changed accordingly (line 336 and in another occurence)

9. Throughout: 1d should be 1D or 1-D

Changed accordingly

10. Throughout: some language is somewhat nonscientific (e.g. spectacular on line 298 is hyperbolic)

Spectacular has been replaced by "substantial" (and the sentence has been rephrased).

We have re-read thoroughly the document and tried, as suggested, to improve some formulations and vocabulary.

**3  Answers to Referee #2**

**spectacular $->$ important ?**

As also noted by Reviewer 1, words as "spectacular" are rather non-scientific. This sentence has been reworded as:

The increase of accuracy and the reduction of diffusion are substantial when . . .

**suggest, describes : plural / singular ?**

As noted our use of plural / siingular for publications with more that one author was inconsistent. The correct use is plural (e.g. Lachatre et al. (2020) describe). This is corrected.

**No extra space after equations**

This has been corrected (except in places in which a paragraph ended, materialized by a dot (.) after the equation. As also highlighted by the reviewer, dots (.) were missing after some equations that end a sentence / paragraph. We have added them where needed throughout. We believe that the Editorial office can help us further in correcting the typographic layout of equations and their environment, shall that be needed.

**No extra space before comma (p. 4)**
This has been corrected.

**l. 126, extra parenthesis**
The extra parenthesis has been removed.

**l. 131, missing dot**
Corrected, thank you.

**l. 194, $w_0 -> W_0$**
Corrected.

**Fig. 1 : should be constant $z$ ?? Check (26)**
As described in lines 207-211 of the manuscript, Eqs. 25-26 describe how the initially rectangular zone containing the tracer is transformed into a tilted parallelogram under the action of wind shear. The figure is in line with Eqs. (25)-(26) and the description in lines 207-211

**4 References**

**References**

Colella, P. and Woodward, P. R.: The piecewise parabolic method (PPM) for gas-dynamical simulations, Journal of Computational Physics, 11, 38–39, 1984.

Després, B. and Lagoutière, F.: Un schma non linéaire anti-dissipatif pour l'équation d'advection linéaire, Comptes Rendus de l'Académie des Sciences - Series I - Mathematics, 328, 939 – 943, https://doi.org/https://doi.org/10.1016/S0764-4442(99)80301-2, URL http://www.sciencedirect.com/science/article/pii/S0764444299803012, 1999.

Godunov, S. K. and Bohachevsky, I.: Finite difference method for numerical computation of discontinuous solutions of the equations of fluid dynamics, Matematičeskij sbornik, 47(89), 271–306, URL https://hal.archives-ouvertes.fr/hal-01620642, 1959.

Lachatre, M., Mailler, S., Menut, L., Turquety, S., Sellitto, P., Guermazi, H., Salerno, G., Caltabiano, T., and Carboni, E.: New strategies for vertical

transport in chemistry-transport models: application to the case of the Mount Etna eruption on March 18, 2012, Geophys. Model Dev. Discuss., 2020.

Van Leer, B.: Towards the ultimate conservative difference scheme. IV. A new approach to numerical convection, Journal of Computational Physics, 23, 276 – 299, https://doi.org/https://doi.org/10.1016/0021-9991(77)90095-X, URL `http://www.sciencedirect.com/science/article/pii/002199917790095X`, 1977.

Zhuang, J., Jacob, D. J., and Eastham, S. D.: The importance of vertical resolution in the free troposphere for modeling intercontinental plumes, Atmospheric Chemistry and Physics, 18, 6039–6055, https://doi.org/10.5194/acp-18-6039-2018, URL `https://www.atmos-chem-phys.net/18/6039/2018/`, 2018.